# Efficient Somatic Embryogenesis, Regeneration and Acclimatization of *Panax ginseng* Meyer: True-to-Type Conformity of Plantlets as Confirmed by ISSR Analysis

**DOI:** 10.3390/plants12061270

**Published:** 2023-03-10

**Authors:** Jung-Woo Lee, Jang-Uk Kim, Kyong-Hwan Bang, Nayeong Kwon, Young-Chang Kim, Ick-Hyun Jo, Young-Doo Park

**Affiliations:** 1Department of Herbal Crop Research, National Institution of Horticultural and Herbal Science, Rural Development Administration, Eumseong 27709, Republic of Korea; 2Department of Horticultural Biotechnology, Kyung Hee University, Yongin 17104, Republic of Korea; 3Research Policy Bureau, Rural Development Administration, Jeonju 54875, Republic of Korea; 4Department of Crop Science and Biotechnology, Dankook University, Cheonan 31116, Republic of Korea

**Keywords:** propagation, in vitro, somatic embryogenesis, regeneration, acclimatization, ISSR, genetic fidelity, *Panax ginseng*

## Abstract

*Panax ginseng* Meyer grows in east Russia and Asia. There is a high demand for this crop due to its medicinal properties. However, its low reproductive efficiency has been a hindrance to the crop’s widespread use. This study aims to establish an efficient regeneration and acclimatization system for the crop. The type of basal media and strength were evaluated for their effects on somatic embryogenesis, germination, and regeneration. The highest rate of somatic embryogenesis was achieved for the basal media MS, N6, and GD, with the optimal nitrogen content (≥35 mM) and NH_4_^+^/NO_3_^−^ ratio (1:2 or 1:4). The full-strength MS medium was the best one for somatic embryo induction. However, the diluted MS medium had a more positive effect on embryo maturation. Additionally, the basal media affected shooting, rooting, and plantlet formation. The germination medium containing 1/2 MS facilitated good shoot development; however, the medium with 1/2 SH yielded outstanding root development. In vitro-grown roots were successfully transferred to soil, and they exhibited a high survival rate (86.3%). Finally, the ISSR marker analysis demonstrated that the regenerated plants were not different from the control. The obtained results provide valuable information for a more efficient micropropagation of various *P. ginseng* cultivars.

## 1. Introduction

*Panax ginseng* Meyer is a medicinal plant that belongs to the Araliaceae family. Ginseng mainly grows in Korea, northeast China, and east Russia. Ginseng root has been widely used as a traditional medicine for thousands of years in east Asia. Physiochemically, this plant contains at least 289 types of triterpene saponins (ginsenosides), which mostly accumulate in the roots [1]. Comprehensive pharmacological and chemical studies have shown that ginsenosides have anti-inflammatory, anti-microbial, anti-oxidative, and cardio-protective properties [2]. Because of these medicinal properties, this species has been overexploited. Therefore, it is necessary to establish efficient propagation methods for the sustainable cultivation and conservation of *P. ginseng*.

Generally, *P. ginseng* propagates through sowing seeds. Seeds are usually produced after a three-year long cultivation; specifically, a plant produces approximately 40 seeds after its juvenile period [3]. Even after seeds have been acquired, stratification for an additional 12–18 months is essential for their germination [4]. Thus, the breeding and propagation of elite lines through conventional methods is a time- and labor-intensive process [5].

Therefore, in vitro culture techniques could offer an alternative means for rapid propagation of uniform crops [6]. Somatic embryogenesis, which is the production of embryos from somatic cells under controlled conditions, represents one of the best ways to produce genotypically valuable cultivars. The value of this approach lies in the fact that it permits a rapid production of numerous plantlets [7,8]. However, the success of in vitro clonal propagation via somatic embryogenesis is contingent upon a variety of factors (e.g., explant type, basal medium composition and strength, and carbon source and concentration) [9]. Additionally, in vitro-grown plants are completely dependent on the mineral nutrients in the culture medium; a supply of an appropriate ratio of minerals through a basal medium is vital for promoting in vitro tissue growth [10]. Regeneration of *P. ginseng* through somatic embryogenesis is well documented within the context of zygotic embryos [11,12], roots [13], and petioles [14]. However, little is known about the influences of basal media and their strengths on somatic embryogenesis. Furthermore, there have been no reports on the optimization of basal media in the germination media used for in vitro culture of *P. ginseng*.

To use in vitro clonal propagation at a commercial scale, it is necessary to determine the conditions for efficient regeneration and acclimatization of ginseng [15]. However, *P. ginseng* is currently difficult to propagate via tissue culture because of its poor survival rate (36%) post-acclimatization [16]. Additionally, within the context of ginseng production, the production of a true-to-type plant—without somaclonal variation—is currently difficult to achieve consistently.

In this study, we established effective regeneration and acclimatization systems for *P. ginseng*. Specifically, optimal basal media and their strengths were tested with regard to their effectiveness in somatic embryogenesis, germination, and plant regeneration. After regenerated plantlets formed in vitro-grown roots (IGRs), they were transferred to soil, and their survival and growth trajectories were investigated. Finally, the genetic fidelity between the regenerated plants and the control was elucidated using ISSR markers.

## 2. Results

### 2.1. Somatic Embryogenesis

Zygotic embryos from naked dehiscent seeds were excised and inoculated in the induction media for somatic embryos (SEs) (Figure 1A–C). After three days of in vitro culture, the colors of the explants started to change from white to green. After 20–25 days of culture, some explants turned red, while the rest remained green (Figure 1D). Regardless of the color of the explants, SEs began to form on the excised surfaces of the explants. This occurred within 30 days after inoculation in the SE induction media. At sixty days after inoculation, SE formation had occurred across all the tested induction media (Appendix A); however, the resultant SEs were significantly different (*p* ≤ 0.001) and these differences were attributed to the type of basal media (Figure 2). The highest SE induction rate was recorded in MS (97.7%), followed by N6 (94.0%), and GD (91.3%). On the other hand, the SE induction rates were relatively low in B5 (72.4%) and SH (65.1%). The number of SEs were highest in MS (23.6), followed by N6 (23.0), GD (22.3), B5 (7.6), and SH (5.4). The embryo-forming capacity (EFC) was highest in MS (23.0), followed by N6 (21.6), GD (20.4), B5 (5.5), and SH (3.6).

Subsequently, the influence of MS strength was studied by supplementing the induction media with four different MS strengths. Although SEs formed in all media tested after 60 days of inoculation (Figure 3), there were significant differences (*p* ≤ 0.001) in somatic embryogenesis; these differences were dependent on the strength of the MS in the induction media (Figure 4). We found that the full MS strength yielded the best SE induction rate (96.9%), followed by 3/2 (88.7%), 1/2 (83.4%), and double strength (59.3%). The maximum number of SEs were observed in the full-strength MS (16.8) and 1/2 MS (16.6) media. With the supplementation of over 3/2 MS strength, the mean number of SEs was significantly lower compared to the full-strength MS medium (*p* ≤ 0.001). The highest EFC index was observed in the full-strength MS medium (16.2), followed by 1/2 MS (13.8), 3/2 MS (9.9), and double strength (5.0).

Explants with immature (globular or heart-shaped) SEs were transferred to the maturation media containing MS of different strengths (1/4, 1/3, 1/2, and full) for subsequent maturation. After four week of culture, SEs of various stages were observed, but their ratio differed based on the MS strength used in the maturation medium (Figure 5). When the diluted MS medium was used as the maturation medium, we found that the ratio of mature SEs (torpedo and cotyledonary) increased. The maximum ratio of mature SEs was observed at 1/3 MS (torpedo 42.5%; cotyledonary 28.5%), followed by 1/4 MS (torpedo 40.4%; cotyledonary 25.0%). On the other hand, high ratio of SEs remained in the immature stage in the maturation medium with full-strength MS.

### 2.2. Plant Regeneration

Four basal media (1/2 GD, 1/2 MS, 1/2 N6, and 1/2 SH) were tested as the germination media for the mature SEs. A high germination rate (above 88%) was observed in all germination media, except for 1/2 GD (67.8%) (Appendix A). However, the basal medium in each of the germination media affected the morphology of both explants and shoots (Figure 6). The most well-developed shoots were obtained from the germination media containing 1/2 MS and 1/2 SH, whereas the shoots produced in the other media were underdeveloped and unhealthy. Specifically, physiological disorders, such as necrosis, and an accumulation of a red pigment were found in the germination media containing 1/2 GD and 1/2 N6.

Induced shoots in each germination medium were then transferred to the elongation medium. This was performed to evaluate the post-effect of the germination media on plant regeneration. The mean number of total regenerated plants was highest in the germination medium containing 1/2 MS (12.2), followed by 1/2 SH (11.0) (Figure 7A). The highest mean number of W types was observed in 1/2 MS (6.6), followed by 1/2 SH (4.9), 1/2 GD (2.8), and 1/2 N6 (2.4) (Figure 7B). The mean number of S types was highest in 1/2 MS (2.7) (Figure 7C), while the maximum number of R types was observed in 1/2 SH (4.8) (Figure 7D). The mean number of plantlets was highest in the germination media containing 1/2 SH (9.7) and 1/2 MS (9.5) (Figure 7E).

The growth characteristics associated with each germination medium were then investigated. The growth of all aerial parts (except their diameter) was greatest in the medium with 1/2 MS (Figure 8 and Appendix A). The growth of underground parts was excellent in plants derived from the medium with 1/2 SH; this growth was measured in terms of the maximum root weight. However, these trends were not significantly different from the others.

### 2.3. Acclimatization

After several months of culture, well-developed in vitro-grown roots (IGRs) derived from somatic embryogenesis were obtained and transferred to soil. Although sprouting occurred within two months in all treatments, the sprouting rate was significantly (*p* ≤ 0.01) driven by the weight of the IGRs (Table 1). The greatest sprouting rates were observed in the 0.60 g group (96.7%). Furthermore, the weight of the IGRs significantly affected (*p* ≤ 0.001) all growth parameters of the aerial parts, as measured via stem diameter, and leaf length and width. When the shoots were senescent and fell off, the roots were harvested and evaluated for survival and growth characteristics (Table 2). We found that as the weight of the IGRs increased, the survival rate gradually improved and was highest in the 0.60 g group (86.3%). Additionally, as the weight of acclimated IGRs increased, underground growth, as measured via root weight, diameter, and length, significantly increased (*p* ≤ 0.001).

### 2.4. Genetic Fidelity

An evaluation of genetic fidelity was performed using the DNA extracted from 11 samples. These samples consisted of 10 sprouted plants from IGRs and one germinated plant as a control. All ISSR primers produced reproducible bands, but their size and number were different (Figure 9). Three primers (UBC818, UBC821, and UBC878) produced only single bands, which were approximately 700 bp. The UBC827 primer produced two bands between 700 and 770 bp. Other primers (UBC809 and UBC868) formed four or more multi-bands, the range of which was between 370 and 770 bp. The DNA bands of both the regenerated plants and the control were monomorphic to each other.

## 3. Discussion

Somatic embryogenesis has been widely used for micropropagation of many species [3,8,17]. However, since this method does not have a universal approach that would enable its implementation across multiple species, optimal approaches need to be individually determined for each species of interest. The primary basis of determining an optimal somatic embryogenesis approach largely entails elucidating the ideal composition of the culture medium [18]. Among many inorganic components that constitute a basal medium, nitrogen plays the most important role in growth and morphogenesis [19]. 

Adil and Jeong [20] found that most protocols for clonal propagation of *P. ginseng* through somatic embryogenesis entailed the use of MS media that were chosen without having been validated. The effect of a basal medium on the formation of SEs of *P. ginseng* has not been studied comprehensively. In this study, the basal medium in the induction media of SEs had an apparent influence on the formation of ginseng SEs (Figure 2 and Appendix A): the potential of somatic embryogenesis was best in the MS medium, followed by the N6 and GD media, and the B5 and SH media were found to be inefficient at inducing SEs of *P. ginseng*. A similar result was reported by Kim et al. [11]; specifically, they found that the MS medium was the best basal medium for SE formation. The greatest factor that differentiated high- and low-efficiency media was nitrogen concentration. Compared to less efficient media (B5 and SH), the most efficient media had the highest nitrogen concentration. The total nitrogen content (60.01 mM) of the MS medium was more than double that of the SH medium (27.34 mM) (Table 3). The total nitrogen content of the N6 and GD media was approximately 35 mM, which was higher than that of the B5 and SH media. Overall, these results suggest that a nitrogen concentration above a specific level (≥35 mM) is essential for efficient somatic embryogenesis of *P. ginseng*. These results are congruent with those of other studies, which have shown that nitrogen concentration in the culture media is the most crucial factor in the induction of SEs [21,22]. During somatic embryogenesis, somatic cells undergo rapid morphological and biochemical changes. These changes demand large quantities of nitrogen, which is needed for the synthesis of large amounts of proteins and nucleic acids [23]. However, the difference in nitrogen concentration could not explain all the differences in the efficiency at which SEs were induced across the different media in this study.

Another factor underlying the differences in the efficiencies of the different basal media were the ratios of different types of nitrogen (i.e., oxidized or reduced). This trend is in line with a previous study, which shows that SE induction and development is not only affected by nitrogen concentration, but also by the type of nitrogen [23]. In the present study, the ratio of NH_4_^+^ to NO_3_^−^ of the most efficient media was 1:2 or 1:4, whereas that of the least efficient media was 1:12 (Table 3). These results indicate that high NH_4_^+^ is required for efficient embryogenic induction of *P. ginseng*. A similar result was reported in a study that showed that NH_4_^+^ stimulated SE formation in *P. ginseng* [11]. Choi and soh [24] reported that SE formation of *P. ginseng* was higher at a ratio of NH^4+^ to NO^3−^ of 1:1 or 1:2 than at other ratios, such as 1:5. Meijer and Brown [25] reported that NH_4_^+^ is necessary for SE induction in *Medicago sativa*. Additionally, an optimal ratio of oxidized and reduced nitrogen promotes morphogenesis and somatic embryogenesis [26]. However, the range of optimal ratios varies by species [27,28]. Our results suggest that the optimal NH_4_^+^ to NO_3_^−^ ratio for the formation of SEs in *P. ginseng* is 1:2 or 1:4.

Determining the optimal strength of culture media is a prerequisite to efficiently performing micropropagation. This is because the strength of the media can significantly impact somatic embryogenesis [29]. In the present study, the MS strength in the medium had a significant effect on somatic embryogenesis (Figure 3 and Figure 4). The full-strength MS medium was the best medium for somatic embryogenesis when compared to other media of other strengths. While only SEs were formed in other MS strengths (Figure 3A–C), callus formation was observed along with SE formation in the double-strength MS medium (Figure 3D). A similar result was reported in a previous study; the authors found that high NH_4_NO_3_ concentrations (over 60 mM) promoted callus formation, while inhibiting somatic embryogenesis of *P. ginseng* [30]. The negative effects of high MS strength on somatic embryogenesis have been reported in *Schisandra chinensis* [17]. It is presumed that excessively high salt may cause high osmotic pressure, ionic toxicity, and imbalance, which may adversely affect somatic embryogenesis [30].

Changes in medium strength are closely linked to fluctuations in the osmotic pressure of the medium, and these fluctuations have been shown to affect the development and maturation of SEs [31]. In this study, the MS strength in the maturation medium affected the development of SEs (Figure 5). When the full-strength MS was used in the maturation media, a large proportion of SEs remained immature. However, the diluted MS media had a more positive effect on SE maturation in *P. ginseng*. Congruent with our findings, Kim et al. [3] showed that SE maturation occurred optimally under low salt concentrations. Prakash and Gurumurthi [32] showed that a full-strength medium yielded the highest number of immature embryos, and in contrast, at 50% dilution, the MS medium yielded a better result in terms of the maturation and germination of SEs of *Eucalyptus camaldulensis*.

The endosperm, which provides a zygotic embryo with various nutrients, including nitrogen and energy, during germination, is not present in SEs. Therefore, SEs are dependent on the nutrients supplied in the germination medium [33]. Mineral nutrients in tissue cultures are important because the inorganic components in the basal medium are responsible for the response to plant regeneration in most plant species [18,26]. Several studies have investigated the effect of GA_3_ (i.e., in the germination medium) on the germination of ginseng SEs [13,34]. However, the effect of basal media remains unclear. In the present study, we did not find a significant difference in germination rate due to the basal media in the germination media, except for 1/2 GD (Appendix A), but the quality of shoots was markedly different. Vigorous shoot formation was observed only in the germination media containing 1/2 SH and 1/2 MS (Figure 6B,D). The shoots originating from the germination media supplemented with 1/2 GD or 1/2 N6 were of poor quality; they exhibited various physiological issues, such as necrosis and an abnormal accumulation of red pigment (Figure 6A,C). The red pigment generated in the N6 and GD media is thought to be the accumulation of incomplete chlorophyll synthesis intermediates. Additionally, the results in Figure 7 show that the type of basal media in the germination media had a post-effect on aspects linked to the plant regeneration of *P. ginseng*; these aspects included shooting, rooting, and plantlet formation. The germination media containing 1/2 MS yielded the highest number of regenerated plants: W and S types (Figure 7A–C). The growth of the aerial parts was also best in plants derived from the 1/2 MS medium (Figure 8 and Appendix A). On the other hand, the number of R types and underground growth were the best in the germination media containing 1/2 SH basal medium (Figure 7D and Appendix A). In line with our findings, a study showed that MS medium positively impacted shoot growth of *P. ginseng*, whereas SH medium stimulated the growth of its roots [35]. Choi et al. [16] reported that most of the S types died without root induction. In the present study, only the W- and R-type plants were converted into plantlets.

The number of plantlets was highest in both germination media containing 1/2 MS and 1/2 SH (Figure 7E). Many studies have highlighted approaches that entail inducing the germination of *P. ginseng* SEs using 1/2 MS or 1/2 SH medium as a basal medium of germination media [11,12,16,30,34,36]. Unlike the study by Carlsson et al. [33], who showed that nitrogen in the germination medium is a key substance that controls the germination of SEs, we found no evidence pointing to the importance of nitrogen in the germination media. Instead, we found that the greatest difference (i.e., in terms of inorganic components) between the good germination media and the poor germination media was Mg^2+^ content (Table 3). With regard to plant growth, magnesium is one of the most important nutrients as it contributes to the occurrence of photosynthesis in chloroplasts [37]. Recently, many studies have delved into the effect of magnesium on plant shoot and root formation, and photosynthesis [38,39]. Magnesium deficiency, which can have severe implications in the early stages of plant growth, not only reduces germination but also impairs seedling establishment [40]. Based on these findings, we can infer that the low Mg^2+^ content in the GD and N6 basal media might have negatively affected the germination and growth of regenerated plants.

Micropropagation studies based on plant tissue culture technology have been conducted across many plant species, but few have been successful at a commercial scale because of the low survival rates that characterize acclimatization. Therefore, high survival rate of regenerated plants during acclimatization is necessary for a successful in vitro propagation [15]. Choi et al. [16] showed that only 23% of regenerated plants survived after acclimatization. However, when these plants were treated with GA_3_ before acclimatization, the survival rate increased to 59.6% [41]. The present study showed that sprouting and survival of IGRs were significantly affected by weight (Table 1 and Table 2). Specifically, sprouting, survival, and aerial and underground growth improved as the weight of IGRs increased. The maximum sprouting rate and survival rate were 96.7% and 86.3%, respectively. Unlike most previous research studies, our findings brought forth a high survival rate after acclimatization. We suspect that IGRs support acclimatization. IGRs have been described as an excellent type of soil transplanting source for successful acclimatization of *P. ginseng* [35,36]. Our findings are more detailed that those of previous studies on acclimatization and provide valuable information with regard to the micropropagation of elite *P. ginseng* lines. In future research, acclimated IGRs will be transferred to the field to check whether seed production is normally performed.

To achieve micropropagation through plant tissue culture at a commercial level, it is essential not only to establish an optimal protocol but also to ensure genetic stability. DNA-based molecular markers are robust tools to evaluate genetic uniformity. ISSR has been frequently used to detect genetic variation of regenerated plants. The wide use of this marker approach is largely because of its cost-effectiveness as well as its high reliability and reproducibility [42]. In this study, six ISSR primers were used to evaluate the genetic fidelity of regenerated plants (Appendix A). These plants were compared to a control sample. The results showed that the primers formed 1–5 scoreable bands, ranging from 370 to 770 bp (Figure 9). All band patterns of the 10 regenerated plants were not significantly different from each other. In addition, all band sizes and patterns between the control and regenerated plants were shown to be monomorphic. This result is indicative of an absence of somaclonal variation during plant regeneration via somatic embryogenesis of *P. ginseng*.

## 4. Materials and Methods

### 4.1. Somatic Embryogenesis

The self-pollinated mature seeds of *P. ginseng* cv. Cheonryang, stratified for six months using a method previously described by Lee et al. [4], were provided by the National Institute of Horticultural and Herbal Science, Rural Development Administration, Republic of Korea. After the seed coats were removed (Figure 1A), the seeds were sterilized following an approach described by Lee at al. [35]. To remove soil and bacteria from the naked seeds, they were soaked in a 2% (*v*/*v*) sodium hypochlorite solution for 5 min and were shaken for 5 min. On a clean bench, a second surface sterilization was carried out for one minute using 70% ethanol. The seeds were rinsed 2–3 times in sterile distilled water. The seeds were once more submerged in 2% (*v*/*v*) sodium hypochlorite solution for 20 min, followed by 4–5 rinses with sterile distilled water. Zygotic embryos were then aseptically excised into several explants (1–2 cm) (Figure 1B).

All the explants were inoculated (i.e., abaxial side down) onto the SE induction media (Figure 1C) and observed for color changes and SE induction after 30 days of culture (Figure 1D). To investigate the effect of the basal media on the induction of SEs, five different media, including B5 [43], GD [44], MS [45], N6 [46], and SH [47], were used (Table 3). Subsequently, the effects of the MS media with different strengths (i.e., 1/2, full, 3/2, and double) on SE induction were investigated. All induction media were supplemented with 30 g/L of sucrose and 8 g/L of agar. Each medium was poured onto Petri dishes (60 × 15 mm). All media had their pH set to 5.7 before autoclaving (at 121 °C and 1.2 kgf·cm^−2^ pressure for 15 min), followed by alteration of the pH to 5.2. Ten explants were placed in a Petri dish containing an SE induction medium, and each treatment had at least five replicates. The SE induction rate and the number of SEs per explant were evaluated after 60 days of inoculation in the following manner:

SE induction rate (%): the number of explants containing SE/total number of explants × 100.

Mean number of SEs per explant: *∑* the number of SEs per explant/the number of explants from which at least one SE was induced.

The embryo-forming capacity (EFC) index was also calculated using the following equations, as described in [19]:

EFC: SE induction rate × mean number of SEs per explant/100

The SEs derived from the induction media mostly remained at the immature stage (Figure 1E). To develop and mature the SEs, they were transferred to the maturation media containing different strengths (1/4, 1/3, 1/2, and full) of MS medium. All maturation media were supplemented with 20 g/L of sucrose and 8 g/L of agar. Each medium was poured onto Petri dishes (90 × 15 mm). Ten explants containing approximately 25 SEs were cultured in each maturation medium, and at least 3 Petri dishes were used per treatment. Each Petri dish was considered a replicate. After 30 days of culture, the SEs were examined and classified into globular, heart-shaped, torpedo, and cotyledonary stage (Figure 1E,F).

### 4.2. Plant Regeneration

After 30 days of subculture with the maturation medium, the SEs that had reached maturation (torpedo and cotyledonary) were transferred to the germination medium for shoot induction. A total of 10 explants containing approximately 50 SEs were implanted in each germination medium composed of different basal medium (1/2 GD, 1/2 MS, 1/2 N6, and 1/2 SH). All germination media were supplemented with 20 g/L of sucrose, 8 g/L of agar, and 10 mg/L of gibberellic acid (GA_3_). Each medium was poured onto Petri dishes (100 × 40 mm). Five Petri dishes were used per treatment, and each Petri dish was considered a replicate. After 30 days, photographs were captured (Figure 1G), and the rate of germination was monitored as the percentage of the number of explants containing shoots to the total number of explants. Twenty shoots derived from each germination medium were transferred to the elongation medium (1/2 SH, 20 g/L sucrose, 5 g/L agar, and 5 g/L activated charcoal) to induce further development, for example, rooting and plantlet formation (Figure 1H). Each experimental unit consisted of 5 Petri dishes as a replicate.

After 60 days of culture in the elongation medium, the regenerated plants were divided into three types based on their morphology. W type: a whole plant consisting of both a shoot and a root; S type: plant with shoot only; and R type: plant with root with rhizome (Figure 10). Plantlets were considered to be the sum of the W and R types. This was because most S types withered prior to root formation even after several months of cultivation. The post-effect of the germination medium on plant regeneration was determined by counting the number of these three types of plants; additionally, measures of growth, such as length, diameter, and weight of shoot and root, were determined using a sample of 10 plants derived from each germination medium. Well-developed plantlets were transferred to the growth medium (1/3 SH medium supplemented with 5 g/L of gelrite and 20 g/L of sucrose) (Figure 1I). When the aerial parts senesced, the IGRs derived from the in vitro culture process via somatic embryogenesis were harvested and then cold stratified (2 °C) in darkness for 30 days (Figure 1J).

### 4.3. Acclimatization

To conduct acclimatization studies, the IGRs were classified into four groups based on their weight (0.15, 0.30, 0.45, and 0.60 g) (Figure 11) and soaked with 25 mg/L of GA_3_ for 1 h. Twenty IGRs for each treatment were transferred to artificial soil, which largely consisted of peat moss and perlite (3:1, *v*/*v*). Sprouting was observed within 4 weeks and monitored after 2 months of transplantation (Figure 1K). Simultaneously, the growth of aerial parts (i.e., in terms of length and diameter) and the fresh weight of shoots were examined. When the shoots were defoliated after 5 months of cultivation, the roots were harvested (Figure 1L), and the survival rate (i.e., the ratio of the number of harvested IGRs to the number of transplanted IGRs), the growth of underground parts (i.e., in terms of length and diameter), and the fresh weight of the roots were determined. The growth data were acquired using a method previously described by Lee et al. [35].

### 4.4. Growth Condition

All cultures, including SEs and regenerated plants, were incubated at 23 °C under a 16-h-light/8-h-dark cycle using white fluorescent light at 24 µmol m^−2^ s^−1^. The seeds and IGRs were either sown or transplanted in a greenhouse maintained at 25 ± 2 °C under natural light and watered once a week to keep the soil moist.

### 4.5. Genetic Fidelity Analysis

To ensure the genetic fidelity of the regenerated plants, 10 sprouted shoots derived from the IGRs were chosen randomly along with the control plants (1-year-old ginsengs originated from seeds). Genomic DNA was extracted and quantified using the method previously described by [35]. For molecular marker analysis, 6 sets of ISSR primers were prepared. (Appendix A). PCR reactions were then performed using 20 ng of genomic DNA, 2× master mix (Bio-rad, Copenhagen, Denmark), 7 μL of double-distilled H_2_O, and 10 pmol of ISSR primer. The reaction and cycling conditions were those already described by Lee et al. [48]. The amplified fragments were then analyzed using the automatic capillary electrophoresis system (Qiagen, Valencia, CA, USA).

### 4.6. Data Analysis

The entire study was conducted in a completely randomized design. The results were analyzed as mean ± standard errors. The obtained data were tabularized and analyzed using analysis of variance (one-way ANOVA). The means were compared using Duncan’s multiple analysis for those that were significant as a result of the analysis of variance, and *p* ≤ 0.05 or less was considered to be significant (R program ver. 4.1.2, The R Foundation for Statistical Computing, Vienna, Austria).

## 5. Conclusions

In conclusion, efficient regeneration via a somatic embryogenesis and acclimatization system of *P. ginseng* has been developed in this study. The type of basal media and the strength suitable for SE induction, maturation, and plant regeneration were established. Additionally, a high survival rate and genetic stability were demonstrated when the regenerated plants were acclimated. Our results provide valuable information for the efficient micropropagation of *P. ginseng* at a commercial scale.

## Figures and Tables

**Figure 1 plants-12-01270-f001:**
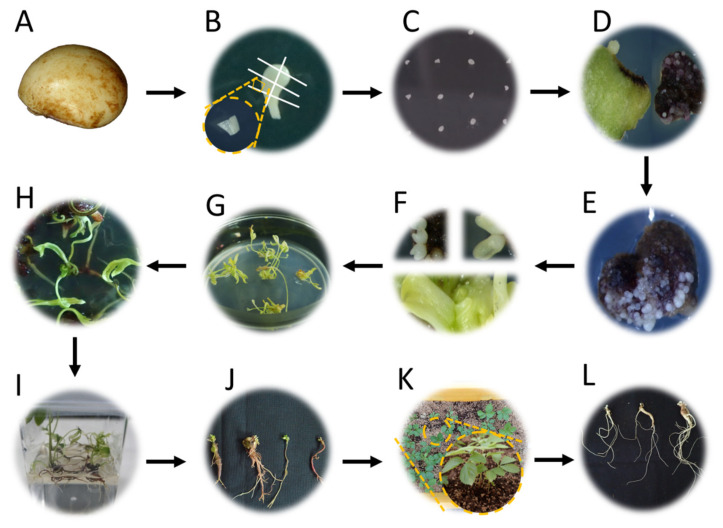
Flowchart of plant regeneration and acclimatization of *P. ginseng*. (**A**) Naked seed after seed coat removal. (**B**) Explants excised from zygotic embryos. (**C**) Transfer to SE induction medium. (**D**) Explants changing from green (left) to red (right) after 30 days of culture. (**E**) Immature (globular stage) SEs after 60 days of inoculation. (**F**) Maturation of SEs: top left: heart-shaped stage; top right: torpedo stage; and bottom: cotyledonary stage. (**G**) Germination of SEs. (**H**) Rooting and development of regenerated plant. (**I**) Plantlets with a well-developed root system. (**J**) IGRs derived from the in vitro culture process via somatic embryogenesis. (**K**) Sprouted IGRs after acclimatization. (**L**) Harvested 2-year-old IGRs. SEs: somatic embryos; IGRs: in vitro-grown roots.

**Figure 2 plants-12-01270-f002:**
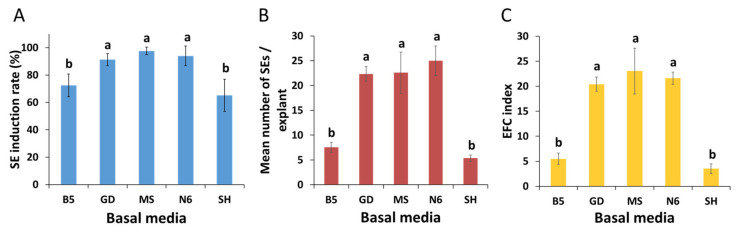
Effects of different basal media on the somatic embryogenesis of *P. ginseng*: (**A**) SE induction rate, (**B**) mean number of SEs per explant, and (**C**) EFC index. The data were recorded 60 days after the inoculation of explants on the SE induction media. The values represent the mean ± standard errors of three independent experiments, each of which consisted of 50 explants. Non-significant or significant differences were determined by an ANOVA test. Different letters within each column represent a significant difference at *p* ≤ 0.05 based on Duncan’s multiple comparison test. SE: somatic embryo; EFC: embryo-forming capacity.

**Figure 3 plants-12-01270-f003:**
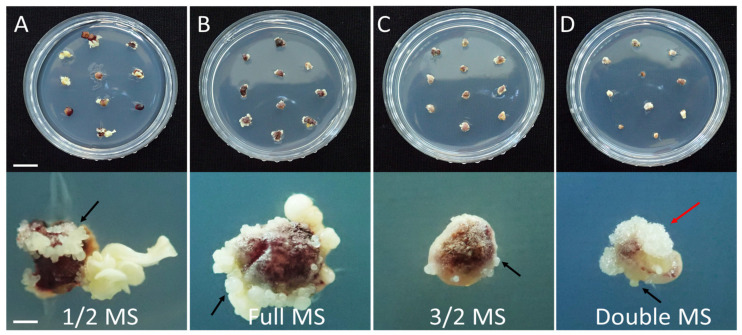
Somatic embryogenesis of *P. ginseng* as influenced by MS strength in the media after 60 days of inoculation: (**A**) 1/2 MS, (**B**) full MS, (**C**) 3/2 MS, and (**D**) double MS. Black arrows and a red arrow indicate the SEs and callus, respectively. Scale bars: upper is 1 cm and bottom is 1 mm. SEs: somatic embryos.

**Figure 4 plants-12-01270-f004:**
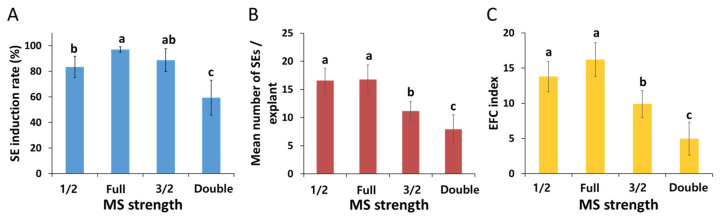
Effects of MS strength on the somatic embryogenesis of *P. ginseng:* (**A**) SE induction rate, (**B**) mean number of SEs per explant, and (**C**) EFC index. The data were recorded 60 days after inoculation of explants on the SE induction media. The values represent the mean ± standard errors of three independent experiments, each of which consisted of 50 explants. Non-significant or significant differences were determined by an ANOVA test. Different letters within each column represent a significant difference at *p* ≤ 0.05 based on Duncan’s multiple comparison test. SEs: somatic embryos, and EFC: embryo-forming capacity.

**Figure 5 plants-12-01270-f005:**
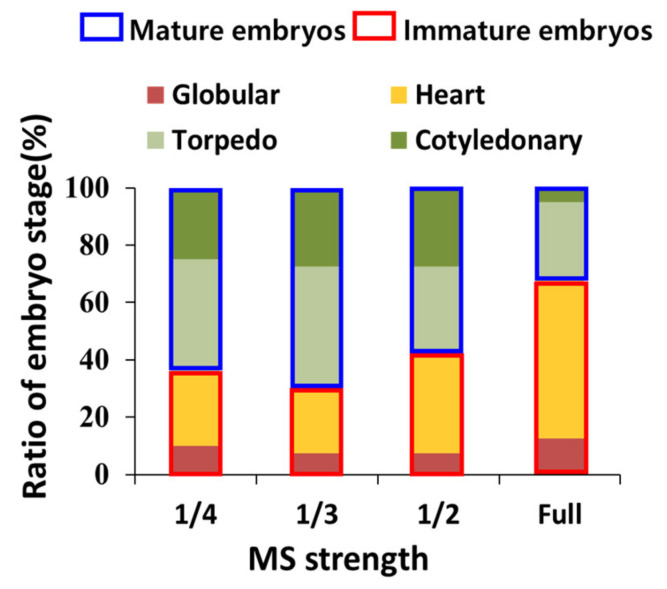
Effect of MS strength on SE maturation of *P. ginseng*. The data were recorded 30 days after the transfer of SEs to the maturation medium. The values represent the ratio of embryo stage of three independent experiments, each of which consisted of 10 explants. Mature embryos: torpedo and cotyledonary stages. Immature embryos: globular and heart-shaped stages. SEs: somatic embryos.

**Figure 6 plants-12-01270-f006:**
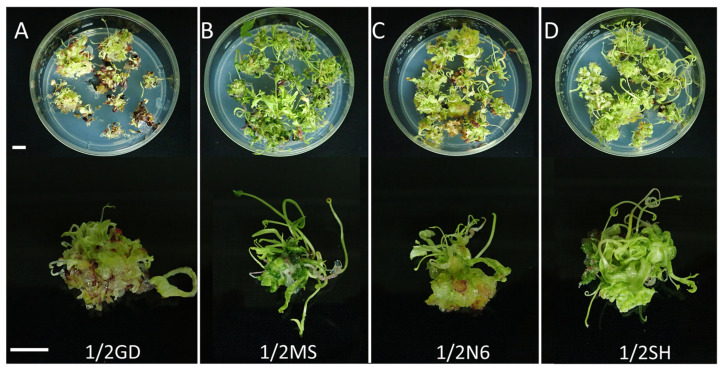
Germination of SEs as influenced by the basal media of the germination media after 30 days of inoculation of *P. ginseng*: (**A**) 1/2 GD, (**B**) 1/2 MS, (**C**) 1/2 N6, and (**D**) 1/2 SH. Scale bars: 1 cm. SEs: somatic embryos.

**Figure 7 plants-12-01270-f007:**
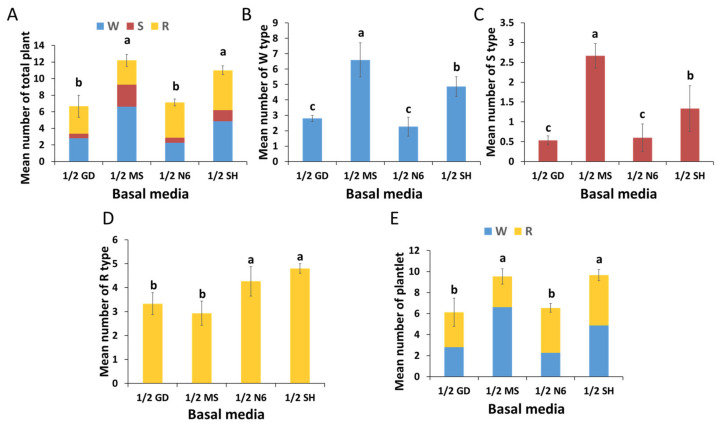
Post-effect of the germination media on plant regeneration: (**A**) mean number of total regenerated plants, (**B**) mean number of W types, (**C**) mean number of S types, (**D**) mean number of R type, and (**E**) mean number of plantlets. The data were recorded 60 days after the transfer of shoots to the elongation medium. The values represent the mean ± standard errors of three independent experiments, each of which consisted of 100 germinated shoots. Non-significant or significant differences were determined by an ANOVA test. Different letters within each column represent significant difference at *p* ≤ 0.05 based on Duncan’s multiple comparison test. **W**: whole plant with shoot and root; **S**: plant with shoot alone; and **R**: plant with root with rhizome.

**Figure 8 plants-12-01270-f008:**
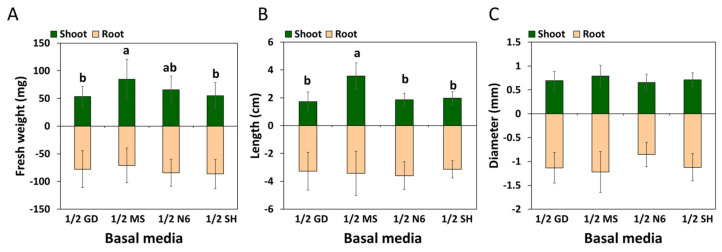
Effect of the germination media on the growth of shoots and roots. The data were recorded 60 days after the transfer of shoots to the elongation medium. Fresh weight, length, and diameter of shoots and roots in various basal media are shown in (**A**–**C**), respectively. The values represent the mean ± standard errors of three independent experiments, each of which consisted of 10 plants. Non-significant or significant differences were determined by an ANOVA test. Different letters within each column represent significant difference at *p* ≤ 0.05 based on Duncan’s multiple comparison test.

**Figure 9 plants-12-01270-f009:**
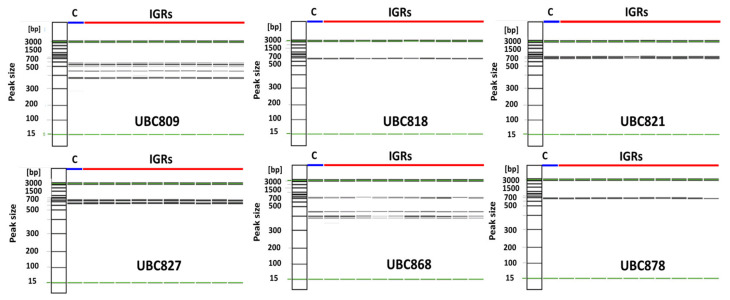
Evaluation of genetic fidelity of regenerated plants using ISSR marker analysis. C: control; IGRs: in vitro-grown roots; and ISSR: inter-simple sequence repeat.

**Figure 10 plants-12-01270-f010:**
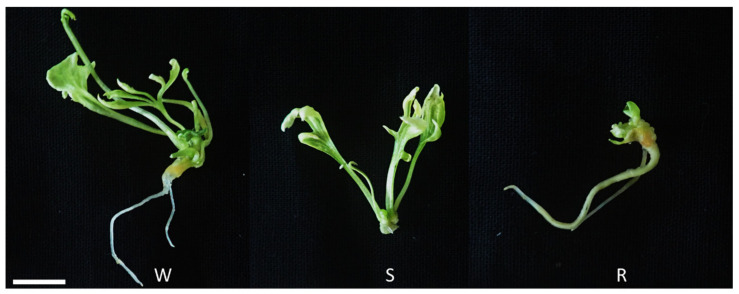
Three types of regenerated plants classified based on morphology. Scale bar: 1 cm. **W**: whole plant consisted of both shoot and root; **S**: plant with shoot only; and **R**: plant with root with rhizome.

**Figure 11 plants-12-01270-f011:**
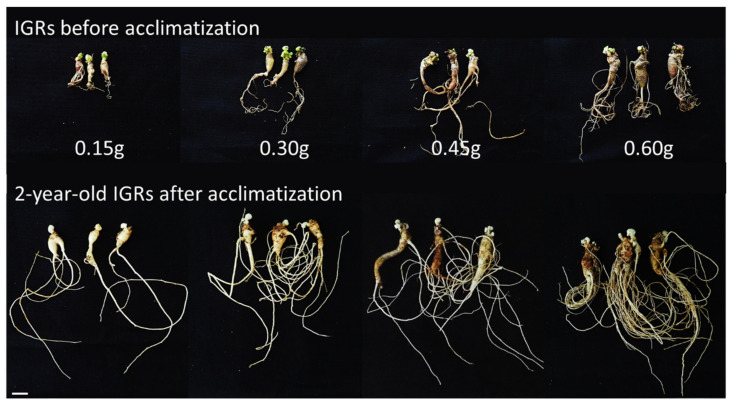
Morphology of IGRs before and after acclimatization. IGRs were classified into four groups by weight (0.15, 0.30, 0.45, and 0.60 g). At the top are IGRs before transplantation to soil. At the bottom are 2-year-old IGRs after acclimatization. Scale bar: 1 cm. IGRs: in vitro-grown roots.

**Table 1 plants-12-01270-t001:** Sprouting and growth of the aerial parts as influenced by the weight of IGRs.

IGR Weight (g)	Sprouting Rate (%)	Stem Length (cm)	Stem Diameter (mm)	Leaf Length (cm)	Leaf Width (cm)
0.15	75.0 ± 8.7 ^b^	4.3 ± 0.3 ^b^	0.88 ± 0.04 ^c^	3.0 ± 0.2 ^d^	1.6 ± 0.2 ^d^
0.30	91.7 ± 2.9 ^a^	4.7 ± 0.5 ^b^	0.93 ± 0.03 ^c^	3.5 ± 0.4 ^c^	2.0 ± 0.2 ^c^
0.45	96.7 ± 5.8 ^a^	5.7 ± 0.4 ^a^	1.03 ± 0.03 ^b^	4.1 ± 0.2 ^b^	2.2 ± 0.1 ^b^
0.60	96.7 ± 2.9 ^a^	6.3 ± 0.7 ^a^	1.18 ± 0.08 ^a^	4.7 ± 0.2 ^a^	2.6 ± 0.1 ^a^

The data were recorded 2 months after the transfer of IGRs to soil. The values represent the mean ± standard errors of three independent experiments, each of which consisted of 20 IGRs. Non-significant or significant differences were determined by an ANOVA test. Different letters within each column represent significant difference at *p* ≤ 0.05 based on Duncan’s multiple comparison test. IGRs: in vitro-grown roots.

**Table 2 plants-12-01270-t002:** Survival and growth of the underground parts as influenced by the weight of IGRs.

IGR Weight (g)	Survival Rate (%)	Root Weight (g)	Root Diameter (mm)	Root Length (cm)
0.15	57.5 ± 15.5 ^b^	0.5 ± 0.0 ^d^	5.28 ± 0.50 ^c^	12.8 ± 0.6 ^b^
0.30	70.0 ± 10.8 ^ab^	0.8 ± 0.2 ^c^	6.45 ± 0.31 ^b^	14.7 ± 1.6 ^b^
0.45	76.3 ± 8.5 ^a^	1.5 ± 0.1 ^b^	7.80 ± 0.55 ^a^	16.8 ± 1.1 ^a^
0.60	86.3 ± 7.5 ^a^	1.8 ± 0.2 ^a^	8.05 ± 0.57 ^a^	17.8 ± 1.6 ^a^

The data were recorded 5 months after the transfer of IGRs to soil. The values represent the mean ± standard errors of three independent experiments, each of which consisted of 20 IGRs. Non-significant or significant difference were determined by an ANOVA test. Different letters within each column represent significant difference at *p* ≤ 0.05 based on Duncan’s multiple comparison test. IGRs: in vitro-grown roots.

**Table 3 plants-12-01270-t003:** Comparison of macro-element ions across the various media used to evaluate somatic embryogenesis and plant regeneration in *P. ginseng*.

	Basal Medium Type				
Macro-Element Ions (mM)	B5	GD	MS	N6	SH
NH_4_^+^	2.02	12.49	20.61	7.00	2.61
NO_3_^−^	24.73	24.46	39.40	27.99	24.73
H_2_PO_4_^−^	1.09	2.2	1.25	2.94	2.61
K^+^	24.73	12.96	20.04	30.93	24.73
Ca_2_^+^	1.02	1.04	2.99	1.13	1.36
Na^+^	1.19	0.10	0.10	0.10	0.05
SO_4_^2−^	2.09	0.15	1.54	4.27	1.62
Cl^−^	2.04	0.87	5.98	2.26	2.72
Mg_2_^+^	1.01	0.14	1.5	0.75	1.62
Total nitrogen	26.75	36.95	60.01	34.99	27.34
NH_4_^+^:NO_3_^−^	1:12	1:2	1:2	1:4	1:12

## Data Availability

All data are included in this paper.

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
