# Peer review of "Efficient Somatic Embryogenesis, Regeneration and Acclimatization of Panax ginseng Meyer: True-to-Type Conformity of Plantlets as Confirmed by ISSR Analysis"

_plants, 2023, doi:10.3390/plants12061270_

Round 1

Reviewer 1 Report

In the submitted manuscript, Jung-Woo Lee et al. evaluated the effect of the basal media type and strength on somatic embryogenesis, germination, and the regeneration of Panax ginseng Meyer,  which provide valuable information for more efficient micropropagation of various P. ginseng cultivars. The experiments are well-designed and provide vital information for P. ginseng generation. My detailed comments are as follows: 

1. The statistical dimension of the Y-axis of all Figures should be deleted. Different letters within each column are enough to represent the significant difference. Similar modifications should be made in the table.

2. EFC index in the corresponding figures should add the statistical difference analysis.

3. The resolution of all figures should be improved. For example, the legends above the figures and the number of the Y axis in Figure 9 are too small.

4. The MS media contained an optimal NH4+ to NO3- ratio (1:2 or 1:4) that was suitable for somatic embryogenesis, hinting that NO3- was more easily absorbed by P. ginseng than NH4+. Did the author detect the change in the expression level of genes involved in  NH4+ and NO3- metabolic pathways, which would strongly support their conclusion?

Author Response

In the submitted manuscript, Jung-Woo Lee et al. evaluated the effect of the basal media type and strength on somatic embryogenesis, germination, and the regeneration of Panax ginseng Meyer,  which provide valuable information for more efficient micropropagation of various P. ginseng cultivars. The experiments are well-designed and provide vital information for P. ginseng generation. My detailed comments are as follows: 

# We appreciate the excellent evaluation and comments on our manuscript, and we have revised the following to reflect the reviewer's opinion.

1. The statistical dimension of the Y-axis of all Figures should be deleted. Different letters within each column are enough to represent the significant difference. Similar modifications should be made in the table.

# Reflecting the reviewer's comment, we revised Figures and Tables.

2. EFC index in the corresponding figures should add the statistical difference analysis.

# Reflecting the reviewer's comment, we added statistical difference analysis of the EFC index.

3. The resolution of all figures should be improved. For example, the legends above the figures and the number of the Y axis in Figure 9 are too small.

# Reflecting the reviewer's comment, we improved the resolution of all figures.

4. The MS media contained an optimal NH4+ to NO3- ratio (1:2 or 1:4) that was suitable for somatic embryogenesis, hinting that NO3- was more easily absorbed by P. ginseng than NH4+. Did the author detect the change in the expression level of genes involved in  NH4+ and NO3- metabolic pathways, which would strongly support their conclusion?

# Thanks for the reviewer's great comment. Unfortunately, we did not consider gene expression analysis related to NH+ and NO3- in this study. We are preparing a transcriptome analysis related to somatic embryogenesis. In future studies, we will perform gene expression analysis related to nitrogen assimilation, including the GS/GOGAT cycle. We apologize for failing to provide nitrogen-related gene expression data. We ask for your understanding.

Reviewer 2 Report

This work presents an interesting descriptive approach for the production of somatic embryos of ginseng. There are several topics that need to be improved for a better understanding and discussion of the technique, also for the proposal of modifications of the somatic embryogenesis for this particular species.

Introduction

- Line 38, how many types of ginsenosides are described in literature? Are some of them in higher concentration in roots, or shoots?

- Line 41, what is the conservation status of this species? Near threatened, vulnerable, endangered, critical?

- Line 47, what is the percentage of germination of the seeds?

- Line 66, what is the percentage of survival of tissue culture of ginseng?

Results

- Paragraph from line 88 to 91, the numbers described are different from what is shown in Figure 2 for MS medium and N6 medium, please check.

- To support Figure 5, if possible, please include some photos of the developing embryos.

- Line 158, the red pigment after treatment with N6 and GD medium could be accumulation of intermediates of incomplete chlorophyll synthesis.

- Line 201, please include the meaning of IGR, “in vitro grown root” since this is the first time it is mentioned.

Discussion

- The paragraph at line 277 talks about the nitrogen and its oxidation or reduced status. Are there any publication about increasing concentration of NH+, in higher proportion than NO3-? Since the plant requires to spend a big amount of energy reducing nitrate into ammonium and amine.

- After that paragraph you can also include information about the concentration of calcium and magnesium in MS medium that are higher than in the other culture media. Calcium for the formation and stabilization of call walls, specially pectin, signal transduction and cell cycle, and magnesium as a cofactor of many enzymes involved in gene expression and other enzymes such as copalyl diphosphate synthase for the synthesis of gibberellins (GAs discussed in paragraph at line 314).

Materials and methods

- Line 390, I would like you to include the details of the sterilization protocol, since this is a delicate procedure that sometimes can damage the embryo inside the seed.

- Line 399, what is the final pH of the culture media after being autoclaved, since frequently the aqueous solutions can get slightly acidified.

- Please include the details of the culture chambers, or the environmental conditions for the seed germination, regeneration, development. Water regime, light intensity and source (natural light, fluorescent, led), temperature, humidity…

Author Response

This work presents an interesting descriptive approach for the production of somatic embryos of ginseng. There are several topics that need to be improved for a better understanding and discussion of the technique, also for the proposal of modifications of the somatic embryogenesis for this particular species.

# We appreciate the excellent comments on our manuscript, and we have revised the following to reflect the reviewer's opinion.

Introduction

- Line 38, how many types of ginsenosides are described in literature? Are some of them in higher concentration in roots, or shoots?

# 289 ginsenosides were observed in Panax genus (Yang et al., 2014). This content has been added to the manuscript with a refererce. (Line 40)

# The type and content of each ginsenoside differ depending on the tissue, organ, and processing method of P. ginseng.

- Line 41, what is the conservation status of this species? Near threatened, vulnerable, endangered, critical?

# P. ginseng can be divided into cultivated and wild ginseng. Cultivated ginseng, including cultivars, are cultivated by farmers, so there is no risk of extinction. However, wild ginseng are native to the mountains and are threatened with extinction due to indiscriminate harvesting. The micropropagation method developed in our study is expected to be useful for wild ginseng preservation.

- Line 47, what is the percentage of germination of the seeds?

# Much research has been performed on ginseng seed germination, and the germination rate is usually 80-90%. Please refer to the previous research result, Korean J Medicinal Crop Sci 24(4): 284 − 293 (2016).

- Line 66, what is the percentage of survival of tissue culture of ginseng?

# Choi et al., (1998) reported a survival rate of 36%. The contents were added to the manuscript. (Line 69)

Results

- Paragraph from line 88 to 91, the numbers described are different from what is shown in Figure 2 for MS medium and N6 medium, please check.

# Reflecting on the reviewer's comment, we revised Figure 2.

- To support Figure 5, if possible, please include some photos of the developing embryos.

# Thanks for the reviewer's great comment. We presented pictures of developing embryos in Figures 1 E, and F, and decided that they could be replaced. We ask for your understanding.

- Line 158, the red pigment after treatment with N6 and GD medium could be accumulation of intermediates of incomplete chlorophyll synthesis.

# Reflecting on the reviewer's opinions, we added sentences in the manuscript (Line 325-326).

- Line 201, please include the meaning of IGR, “in vitro grown root” since this is the first time it is mentioned.

# Reflecting on the reviewer's opinions, we added words and explanations in the manuscript (Line 199-200).

Discussion

- The paragraph at line 277 talks about the nitrogen and its oxidation or reduced status. Are there any publication about increasing concentration of NH+, in higher proportion than NO3-? Since the plant requires to spend a big amount of energy reducing nitrate into ammonium and amine.

# Similar to our findings, many studies have indicated that nitrate is beneficial to somatic embryogenesis at a higher ratio than ammonium. Choi and soh (1997) reported that the somatic embryo formation of P. ginseng was higher at the NH4+ to NO3- the ratio of 1:1 or 1:2 than at other ratios such as 1:5. This content and reference have been added to the manuscript. (Line 281-283)

- After that paragraph you can also include information about the concentration of calcium and magnesium in MS medium that are higher than in the other culture media. Calcium for the formation and stabilization of call walls, specially pectin, signal transduction and cell cycle, and magnesium as a cofactor of many enzymes involved in gene expression and other enzymes such as copalyl diphosphate synthase for the synthesis of gibberellins (GAs discussed in paragraph at line 314).

# Thanks for the reviewer's great comment. Regarding magnesium, we have already discussed this in Line 344-353.

# The reviewer's comments about calcium were quite noteworthy. However, there was no difference in calcium concentration between the basal media except MS media. For this reason, we apologize for not being able to add the reviewer's comments to this manuscript. We ask for your understanding.

Materials and methods

- Line 390, I would like you to include the details of the sterilization protocol, since this is a delicate procedure that sometimes can damage the embryo inside the seed.

# Reflecting on the reviewer's comment, we added sterilization steps to the manuscript. (Line 389-394)  

- Line 399, what is the final pH of the culture media after being autoclaved, since frequently the aqueous solutions can get slightly acidified.

# Reflecting on the reviewer's comment, we added the final pH data after autoclaving. (Line 402-404)  

- Please include the details of the culture chambers, or the environmental conditions for the seed germination, regeneration, development. Water regime, light intensity and source (natural light, fluorescent, led), temperature, humidity…

# Reflecting on the reviewer's comment, we added growth condition of cultures, seeds, and IGRs to the manuscript. (Line 474-478)  

Reviewer 3 Report

Dear Editor, 

The work presented by Lee et al., is interesting  

1.      What is the main question addressed by the research?

This research addresses efficient somatic embryogenesis, regeneration and acclimatization of Panax ginseng Meyer: true-to-type conformity of plantlets by ISSR analysis

2. Do you consider the topic original or relevant in the field? Does it

address a specific gap in the field?

yes

3. What does it add to the subject area compared with other published?

material?

It is distinctively presented in its form

4. What specific improvements should the authors consider regarding the

methodology? What further controls should be considered?

 Figure quality improvement

5. Are the conclusions consistent with the evidence and arguments presented?

and do they address the main question posed?

In my opinion yes

6. Are the references appropriate?

yes

7. Please include any additional comments on the tables and figures.

 None

Author Response

The work presented by Lee et al., is interesting  

  1. What is the main question addressed by the research?

This research addresses efficient somatic embryogenesis, regeneration and acclimatization of Panax ginseng Meyer: true-to-type conformity of plantlets by ISSR analysis

  1. Do you consider the topic original or relevant in the field? Does it

address a specific gap in the field? Yes

  1. What does it add to the subject area compared with other published?

material? It is distinctively presented in its form

  1. What specific improvements should the authors consider regarding the

methodology? What further controls should be considered?  Figure quality improvement

  1. Are the conclusions consistent with the evidence and arguments presented? and do they address the main question posed? In my opinion yes
  2. Are the references appropriate?

Yes

  1. Please include any additional comments on the tables and figures.

 None

# We appreciate the excellent comments on our manuscript.

Reviewer 4 Report

Comments and Suggestions for Authors

The manuscript submitted for review was aimed to establish an efficient regeneration and acclimatization system for Panax ginseng Meyer. a medicinal plant from Araliaceae family growing mainly in Korea, north-east China, and east Russia. The ginseng root has been widely used as a traditional medicine for thousands of years in east Asia. The low reproductive efficiency has been a hindrance to its widespread use as a crop. The somatic embryogenesis were the techic used for regeneration of P. ginseng in the present study accentuating the influences of basal media and their strengths on somatic embryogenesis for which up to now little is known about. Because off poor survival rate  post-acclimatization of the plantlets in ginseng in vitro cultures, the survival and growth trajectories were also investigated, and the genetic fidelity between the regenerated plants and the control was elucidated using ISSR markers.

The manuscript is structured according the requirements of “Plants”. The used methods are appropriate, comprehensive and sufficient to achieve the objectives of the study. The results are presented and illustrated with representatives tables and figures.

However, I have some comments and suggestions regarding the style of the manuscript and some inaccuracies in the terms used:

Abstract:

Somatic embryogenesis was best in the MS, N6, and GD basal media, which contained nitrogen (≥ 35 mM) and an optimal NH4+ to NO3- ratio (1:2 or 1:4).” –my suggestion:  “The highest rate of somatic embryogenesis was achieved on basal media MS, N6 and GD, with optimal nitrogen content (≥ 35 mM) and NH4+ / NO3- ratio (1:2 or 1:4)."

 “These results will provide valuable information for a more efficient micropropagation of various P. ginseng cultivars.” – my suggestion: “The obtained results provide valuable information for a more efficient micropropagation of various P. ginseng cultivars.”

“Full-strength MS medium was the best basal medium strength for somatic embryo induction.” my suggestion:- Full-strength MS medium was the best one for somatic embryo induction.

“However, the diluted MS medium had a more positive effect on embryonic maturation.” – “embryo maturation” instead of “embryonic maturation”

Materials and Methods:

Line 389-390 : “..the seeds were sterilized following an approach described Lee at al. [33]” is omitted “by”/described by Lee at al/

Line 401-402:” The SEs 401 induction rate and the number of SEs per explant were investigated after 60 days ..” “evaluated” is more appropriate instead of “investigated”

Results:

Line 86: “The SE induction rate..” have to be “The highest SE induction rate..”

Line 151-152: “Mature SEs were planted to germination media containing four basal media (1/2 GD, 151 1/2 MS, 1/2 N6, 1/2 SH). – my suggestion:  “Four basal media (1/2 GD, 151 1/2 MS, 1/2 N6, 1/2 SH) were tested as a germination media for the mature SEs

Discussion

Line 253-257: “In this study, the basal medium in the induction media of the SEs had an apparent influence on the formation of ginseng SEs formation (Figure 2 and Figure S1). The potential of somatic embryogenesis was best in MS medium, followed by N6 and GD media. Compared to the other media (MS, N6, GD), B5 and SH were found to be inefficient  at inducing the SEs of P. ginseng.” - my suggestion: “In this study, the basal medium in the induction media of the SEs had an apparent influence on the formation of ginseng SEs formation (Figure 2 and Figure S1): the potential of somatic embryogenesis was best in MS medium, followed by N6 and GD media, and B5 and SH were found to be inefficient at inducing the SEs of P. ginseng.”

In general, the entire text should be carefully reviewed for stylistic and technical inaccuracies.

In conclusion, this manuscript is recommended for publication in “Plants”.

Author Response

Comments and Suggestions for Authors

The manuscript submitted for review was aimed to establish an efficient regeneration and acclimatization system for Panax ginseng Meyer. a medicinal plant from Araliaceae family growing mainly in Korea, north-east China, and east Russia. The ginseng root has been widely used as a traditional medicine for thousands of years in east Asia. The low reproductive efficiency has been a hindrance to its widespread use as a crop. The somatic embryogenesis were the techic used for regeneration of P. ginseng in the present study accentuating the influences of basal media and their strengths on somatic embryogenesis for which up to now little is known about. Because off poor survival rate  post-acclimatization of the plantlets in ginseng in vitro cultures, the survival and growth trajectories were also investigated, and the genetic fidelity between the regenerated plants and the control was elucidated using ISSR markers.

The manuscript is structured according the requirements of “Plants”. The used methods are appropriate, comprehensive and sufficient to achieve the objectives of the study. The results are presented and illustrated with representatives tables and figures.

However, I have some comments and suggestions regarding the style of the manuscript and some inaccuracies in the terms used:

# We appreciate the excellent evaluation and comments on our manuscript, and we have revised the following to reflect the reviewer's opinion.

Abstract:

“Somatic embryogenesis was best in the MS, N6, and GD basal media, which contained nitrogen (≥ 35 mM) and an optimal NH4+ to NO3- ratio (1:2 or 1:4).” –my suggestion:  “The highest rate of somatic embryogenesis was achieved on basal media MS, N6 and GD, with optimal nitrogen content (≥ 35 mM) and NH4+ / NO3- ratio (1:2 or 1:4)."

# Thanks for the reviewer's great comment. Reflecting on the reviewer's comment, we revised our manuscript. (Line 23-25)

 “These results will provide valuable information for a more efficient micropropagation of various P. ginseng cultivars.” – my suggestion: “The obtained results provide valuable information for a more efficient micropropagation of various P. ginseng cultivars.”

# Reflecting on the reviewer's comment, we revised our manuscript. (Line 31-32)

“Full-strength MS medium was the best basal medium strength for somatic embryo induction.” my suggestion:- „Full-strength MS medium was the best one for somatic embryo induction.

# Reflecting on the reviewer's comment, we revised our manuscript. (Line 25)

“However, the diluted MS medium had a more positive effect on embryonic maturation.” – “embryo maturation” instead of “embryonic maturation”

# Reflecting on the reviewer's comment, we revised our manuscript. (Line 26)

Materials and Methods:

Line 389-390 : “..the seeds were sterilized following an approach described Lee at al. [33]” is omitted “by”/described by Lee at al/

# Reflecting on the reviewer's comment, we revised our manuscript. (Line 389)

Line 401-402:” The SEs induction rate and the number of SEs per explant were investigated after 60 days ..” “evaluated” is more appropriate instead of “investigated”

# Reflecting on the reviewer's comment, we revised our manuscript. (Line 406)

Results:

Line 86: “The SE induction rate..” have to be “The highest SE induction rate..”

# Reflecting on the reviewer's comment, we revised our manuscript. (Line 88-89)

Line 151-152: “Mature SEs were planted to germination media containing four basal media (1/2 GD, 151 1/2 MS, 1/2 N6, 1/2 SH). – my suggestion:  “Four basal media (1/2 GD, 1/2 MS, 1/2 N6, 1/2 SH) were tested as a germination media for the mature SEs

# Reflecting on the reviewer's comment, we revised our manuscript. (Line 151-152)

Discussion

Line 253-257: “In this study, the basal medium in the induction media of the SEs had an apparent influence on the formation of ginseng SEs formation (Figure 2 and Figure S1). The potential of somatic embryogenesis was best in MS medium, followed by N6 and GD media. Compared to the other media (MS, N6, GD), B5 and SH were found to be inefficient  at inducing the SEs of P. ginseng.” - my suggestion: “In this study, the basal medium in the induction media of the SEs had an apparent influence on the formation of ginseng SEs formation (Figure 2 and Figure S1): the potential of somatic embryogenesis was best in MS medium, followed by N6 and GD media, and B5 and SH were found to be inefficient at inducing the SEs of P. ginseng.”

# Reflecting on the reviewer's comment, we revised our manuscript. (Line 251-255)

In general, the entire text should be carefully reviewed for stylistic and technical inaccuracies.

In conclusion, this manuscript is recommended for publication in “Plants”.

# We appreciate the excellent evaluation and comments on our manuscript.

Round 2

Reviewer 2 Report

The authors included several of the suggested corrections, included new information and improved many topics declared in the manuscript.

I endorse the publishing of this new version of the manuscript.